# Transcatheter Arterial Embolization for Spontaneous Hepatic Rupture Associated with HELLP Syndrome: A Case Report

**DOI:** 10.3390/medicina57101055

**Published:** 2021-10-02

**Authors:** In-Chul Nam, Jung-Ho Won, Sungbin Kim, Kyungsoo Bae, Kyung-Nyeo Jeon, Jin-Il Moon, Eun Cho, Ji-Eun Park, Jae-Yool Jang, Sung-Eun Park

**Affiliations:** 1Department of Radiology, Gyeongsang National University School of Medicine and Gyeongsang National University Changwon Hospital, Changwon 51472, Korea; sky_hall@naver.com (I.-C.N.); kmsgbn0510@gmail.com (S.K.); ksbae@gnu.ac.kr (K.B.); knjeon@gnu.ac.kr (K.-N.J.); drlotus@naver.com (J.-I.M.); sgeisilver@naver.com (E.C.); 2Department of Radiology, Gyeongsang National University School of Medicine and Gyeongsang National University Hospital, Jinju 52727, Korea; circlehoya@naver.com; 3Department of Obstetrics and Gynecology, Gyeongsang National University School of Medicine and Gyeongsang National University Changwon Hospital, Changwon 51472, Korea; jl1104@daum.net; 4Department of General Surgery, Gyeongsang National University School of Medicine and Gyeongsang National University Changwon Hospital, Changwon 51472, Korea; alitaalita@naver.com

**Keywords:** HELLP syndrome, spontaneous hepatic rupture, angioembolization

## Abstract

*Background:* Spontaneous hepatic rupture associated with the syndrome characterized by hemolysis, elevated liver enzymes, and a low platelet count (HELLP syndrome) is a rare and life-threatening condition, and only a few cases regarding the management of this condition through transcatheter arterial embolization (TAE) have been previously reported. *Case summary:* Herein, we report a case involving a 35-year-old pregnant woman who presented at 28 weeks of gestation with right upper quadrant pain, hypotension, and elevated levels of liver enzymes. Transabdominal ultrasound revealed fetal death. She required an emergency cesarean section, and hepatic rupture was identified after the fetus had been delivered. Hepatic packing and TAE were performed. The postprocedural course was uneventful, and the patient was discharged 14 days after she had been admitted to our hospital. *Conclusions:* Spontaneous hepatic rupture associated with HELLP syndrome is a very serious condition that requires prompt and decisive management. The high maternal and fetal mortality rates associated with this condition can be reduced through early accurate diagnosis and adequate management. The findings in the reported case indicate that TAE may be an attractive alternative to surgery for the management of spontaneous hepatic rupture associated with HELLP syndrome.

## 1. Introduction

Spontaneous hepatic rupture is a rare but life-threatening complication of preeclampsia that is frequently associated with the syndrome characterized by hemolysis, elevated liver enzymes, and a low platelet count (HELLP syndrome). The incidence of spontaneous hepatic rupture that occurs during pregnancy is reported to be between 1 in 45,000 and 1 in 225,000 pregnancies [1].

The most common clinical signs of hepatic rupture are right upper quadrant (RUQ) pain, epigastric pain, severe right shoulder pain, nausea, vomiting, abdominal distention, and hypovolemic shock. For patients with such symptoms, the diagnosis of this condition is delayed because of the non-specific nature of the symptoms. However, epigastric and RUQ pain are considered the most alarming symptoms of this syndrome. It is assumed that these symptoms are caused by the stretching of Glisson’s capsule due to a sinusoidal obstruction of blood flow [2].

The diagnosis of a subcapsular hematoma associated with HELLP syndrome is often delayed because of the variable presentation and low incidence of the condition. The cornerstones of the diagnosis of this condition include clinical examination, laboratory findings, and hepatic imaging. If hepatic involvement is suspected, hepatic imaging with transabdominal ultrasound (TAUS), computed tomography (CT), or magnetic resonance imaging can be performed. However, in most cases, subcapsular hematoma is identified through surgical findings associated with severe maternal and fetal distress [3].

Over the past few decades, there has been a dramatic change in the treatment of hepatic rupture, with a shift toward endovascular treatment with transcatheter arterial embolization (TAE). In the cases of hepatic rupture, achieving liver hemostasis through surgery is difficult because of the presence of multiple areas that are affected by infarction and hematomas and especially because of coagulopathy. Therefore, endovascular treatment is useful for a therapeutic blockade of the hepatic artery [4].

In this report, we describe a case in which a primigravida woman with spontaneous hepatic rupture associated with HELLP syndrome was successfully treated with TAE. A high index of suspicion and prompt recognition are crucial for the purposes of making a proper diagnosis and managing this life-threatening condition. A proper approach involves immediate intervention and the provision of hemodynamic support. 

## 2. Case Presentation

This study was approved by Gyeongsang National University Changwon Hospital Institutional Review Board and informed consent was waived by the Institutional Review Board. 

A 35-year-old primigravida woman without a history of any relevant conditions was admitted to a local hospital at 28 weeks of gestation. She reported RUQ pain and vomiting and received basic medical care. She was referred to our institution because it was suspected that she had acute cholecystitis with septic shock. Laboratory tests revealed thrombocytopenia with a platelet count of 49,000/µL, elevated transaminase levels (AST level of 613 U/L and an ALT level of 533 U/L), and an elevated lactic acid dehydrogenase level (1020 U/L). Her blood pressure (BP) was 70/40 mmHg, and a low hemoglobin level (9.0 g/dL) indicated that she had anemia. TAUS performed after admission revealed fetal death.

Emergency cesarean section was immediately performed as opposed to vaginal delivery to rapidly deliver a dead fetus and to check for accompanying bleeding. When the peritoneum was opened, we encountered an unexpected gush of blood, hence, we suspected that she had concomitant HELLP syndrome. After the stillborn fetus had been quickly delivered and the uterus had been sutured, we confirmed the finding of placental abruption. We explored the abdominal cavity to identify the source of bleeding. The site of hepatic rupture in the right lobe with a subcapsular hematoma was identified and packed. Four hours after surgery, the patient had a distended abdomen and hypovolemic shock, with a BP of 65/40 mmHg, heart rate of 120 beats/min, and urine output of 15 mL/h. TAUS revealed massive hemoperitoneum and a subcapsular hematoma. We suspected that the patient had persistent bleeding, and consequently, digital subtraction angiography was performed under local anesthesia. Selective hepatic angiography revealed contrast media extravasation in the distal branch of the right hepatic artery (Figure 1). 

TAE was successfully performed using the isolation technique (Figure 2). We also performed uterine artery embolization to control the concomitant DIC-induced postpartum bleeding. The patient’s condition stabilized after endovascular treatment. Through blood transfusion, the patient received eight units of packed red blood cells, three units of fresh frozen plasma, and nine units of pooled thrombocytes. Four weeks after the TAE, her liver-function test results and platelet count returned to normal. CT performed after the patient had been discharged revealed the early organization of a subcapsular hepatic hematoma. Except for this observation, no significant changes were observed (Figure 3).

## 3. Discussion

The occurrence of spontaneous hepatic rupture during pregnancy is a rare but potentially fatal event. It is almost always associated with preeclampsia and HELLP syndrome and carries an increased risk of maternal and fetal morbidity and mortality. Maternal mortality has been reported to be as high as 17–59% [5]. Although the exact mechanism leading to hepatic rupture is not known, the occurrence of hepatic rupture has been attributed to periportal hemorrhage and intravascular fibrin aggregation which cause sinusoidal obstruction, intrahepatic vascular congestion, increased hepatic pressure, hepatic necrosis, and intraparenchymal and subcapsular hemorrhage, which may result in capsular rupture. It has been reported that hepatic rupture occurs most often in the right lobe of the liver [6]. The symptoms of spontaneous hepatic rupture, including nausea, vomiting, epigastric or RUQ pain, and hypotension, can be nonspecific. As in the present case, these symptoms could mistakenly be considered symptoms associated with acute cholecystitis or pancreatitis. TAUS is a useful diagnostic method for differentiating hepatic rupture from other suspected diseases.

The standard management of patients with HELLP syndrome who are suspected of having hepatic rupture involves the performance of emergency cesarean section followed by immediate exploratory laparotomy. At this point, different procedures, including procedures involving the use of a dedicated hemostasis device with hepatorrhaphy, perihepatic packing, and hepatectomy, could be performed. In the case described here, because of the amount of blood loss, the difficulty of achieving hemostasis, and the necessity of performing an emergency cesarean section, which added even more time to the procedure, it was evident that appropriate action would be minimal with only perihepatic packing. Unfortunately, bleeding could not be stopped through only packing. In such situations, it is possible that a second-look surgery can be adopted quickly. However, we decided that besides a second-look surgery, the performance of an additional procedure was needed. Therefore, for a complementary approach, TAE was performed. With this additional management, bleeding was controlled, and the gauze pads used for perihepatic packing were removed after the patient’s condition had been stabilized. No additional bleeding was observed at the time.

Conservative nonoperative management for highly selected patients has been recommended [5]. Nevertheless, the standard procedure requires the induction of labor, even for cesarean section, if necessary. Although conservative management of patients with an isolated subcapsular hepatic hematoma after trauma is well established in the literature, we believe that conservative management is inapplicable for patients in whom the same incident occurs during pregnancy, along with bleeding and DIC that arises due to thrombocytopenia. Thus, for first-line treatment in refractory cases, we recommend the use of aggressive intervention in the form of an emergency cesarean section with TAE or exploratory laparotomy. 

Transcatheter embolization of the hepatic artery for hepatic rupture associated with trauma or malignancy is a well-recognized treatment method. Nevertheless, endovascular management of spontaneous hepatic rupture associated with HELLP syndrome has been reported in few papers and referred to as a minor optional treatment in certain review articles [5,7,8,9]. Although TAE is widely used for the treatment of hepatic rupture, it appears that the occurrence of spontaneous hepatic rupture associated with HELLP syndrome is so rare that the use of TAE for the treatment of this condition is not well established. For the treatment of both hepatic rupture and DIC-induced postpartum hemorrhage, TAE is useful and less invasive than other techniques [10]. Therefore, we recommend the use of TAE as a bridge between perihepatic packing and hepatectomy for the management of spontaneous hepatic rupture associated with HELLP syndrome. Formal hepatectomy is an option to be considered when bleeding cannot be stopped through TAE or when there is extensive hepatic necrosis due to the risk of infection [9].

## 4. Conclusions

For pregnant patients who present with a sudden onset of epigastric and/or right quadrant pain accompanied by manifestations of early hemodynamic shock, the possible occurrence of spontaneous hepatic rupture should always be considered. For patients with spontaneous hepatic rupture associated with HELLP syndrome and hemodynamic instability, TAE, which is a technique that can be used for successful and life-saving management through the stabilization and control of acute bleeding, represents an alternative to surgery.

## Figures and Tables

**Figure 1 medicina-57-01055-f001:**
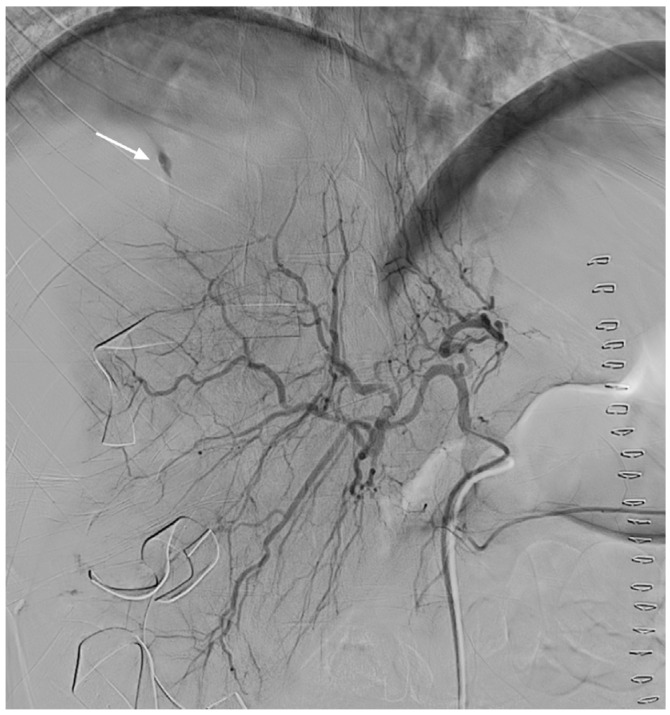
Selective hepatic angiography shows some contrast media extravasation (arrow) in the distal branch of the right hepatic artery.

**Figure 2 medicina-57-01055-f002:**
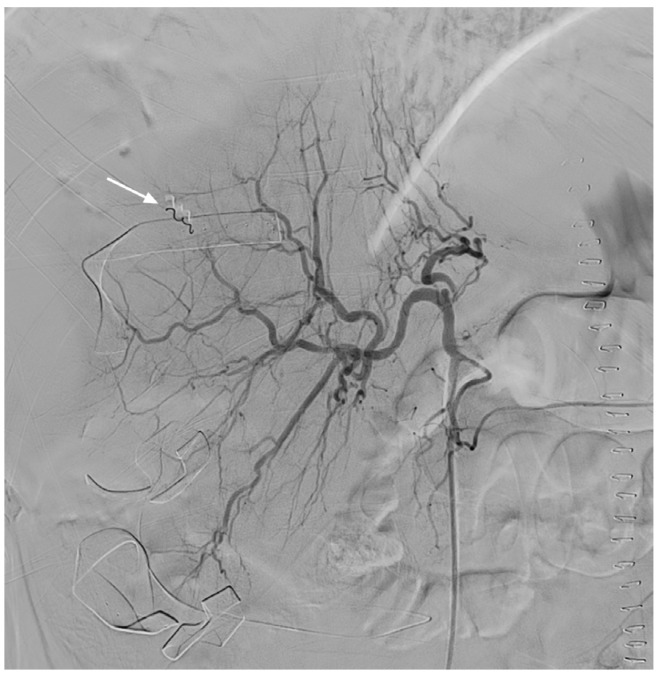
Transcatheter arterial coil embolization (arrow) is successfully performed using the isolation technique.

**Figure 3 medicina-57-01055-f003:**
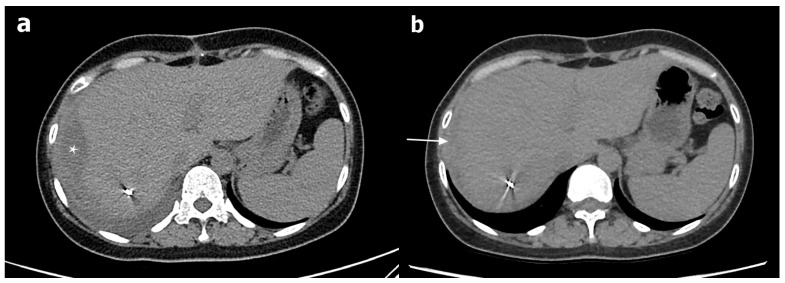
(**a**) Computed tomography (CT) performed four weeks after the TAE reveals the early organization of a subcapsular hepatic hematoma (asterisk). (**b**) CT performed six weeks after the TAE reveals the near-complete resolution of the preexisting subcapsular hematoma (arrow).

## Data Availability

The anonymized data that support the results of this study are available and shared on reasonable request by any qualified investigator.

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
