# Peer review of "Transcatheter Arterial Embolization for Spontaneous Hepatic Rupture Associated with HELLP Syndrome: A Case Report"

_medicina, 2021, doi:10.3390/medicina57101055_

Round 1
Reviewer 1 Report
The authors have significantly revised the manuscript. This is a very informative and well-written case report.
Reviewer 2 Report
No comments
This manuscript is a resubmission of an earlier submission. The following is a list of the peer review reports and author responses from that submission.
Round 1
Reviewer 1 Report
This is a very instructive and well-written case report, and deserves to be published. In order to make it a better paper, I suggest the following modifications.
- The “Introduction” session is too long. Most of the content should be included in the "Discussion" section. The introduction is just an appetizer, so to speak. If readers are full from the appetizer alone, they will not be able to taste the main dish.
- In the end, it was the right decision, but why did authors choose to have a C-section instead of vaginal delivery?
- Was the anesthesia a general anesthesia?
- Did authors noticed the intraabdominal bleeding by pre-operative transabdominal ultrasound?
- Please describe the possible cause of the fetal death.
- Are there any pathological examination of placenta? Microscopical findings of placenta would be informative.
Author Response
Reviewer #1:
1) The “Introduction” session is too long. Most of the content should be included in the "Discussion" section. The introduction is just an appetizer, so to speak. If readers are full from the appetizer alone, they will not be able to taste the main dish.
Response: We would like to thank you for your valuable comment and agree with your suggestion. We are happy to inform you that we have now simplified the introduction in our revised manuscript to increase readability.
Reviewer 2 Report
I suggest you to change the description of the time window. Instead of: "....weeks after the patient had been discharged..." I suggest: ".... weeks after the TAE...". Change everywhere in the text.
Author Response
Reviewer #2:
1) I suggest you to change the description of the time window. Instead of: "....weeks after the patient had been discharged..." I suggest: ".... weeks after the TAE...". Change everywhere in the text.
Response: We would like to thank you for your valuable comment. We are happy to inform you that we have now changed the descriptions of the time window as suggested (Lines: 116,117, 126-129).